# A Novel High-Frequency Vibration Error Estimation and Compensation Algorithm for THz-SAR Imaging Based on Local FrFT

**DOI:** 10.3390/s20092669

**Published:** 2020-05-07

**Authors:** Yinwei Li, Li Ding, Qibin Zheng, Yiming Zhu, Jialian Sheng

**Affiliations:** 1Terahertz Technology Innovation Research Institute, University of Shanghai for Science and Technology, Shanghai 200093, China; liyw@usst.edu.cn (Y.L.); sunnylding@usst.edu.cn (L.D.); qbzheng@usst.edu.cn (Q.Z.); 2Shanghai Key Laboratory of Modern Optical System, University of Shanghai for Science and Technology, Shanghai 200093, China; 3Shanghai Institute of Intelligent Science and Technology, Tongji University, Shanghai 200092, China; 4Shanghai Radio Equipment Research Institute, Shanghai 201109, China; SJL_Jialian@163.com

**Keywords:** terahertz synthetic aperture radar (THz-SAR) imaging, high-frequency vibration error, local fractional Fourier transform (LFrFT), sinusoidal frequency modulation (SFM)

## Abstract

Compared with microwave synthetic aperture radar (SAR), terahertz SAR (THz-SAR) is easier to achieve ultrahigh-resolution image due to its higher frequency and shorter wavelength. However, higher carrier frequency makes THz-SAR image quality very sensitive to high-frequency vibration error of motion platform. Therefore, this paper proposes a novel high-frequency vibration error estimation and compensation algorithm for THz-SAR imaging based on local fractional Fourier transform (LFrFT). Firstly, the high-frequency vibration error of the motion platform is modeled as a simple harmonic motion and THz-SAR echo signal received in each range pixel can be considered as a sinusoidal frequency modulation (SFM) signal. A novel algorithm for the parameter estimation of the SFM signal based on LFrFT is proposed. The instantaneous chirp rate of the SFM signal is estimated by determining the matched order of LFrFT in a sliding small-time window and the vibration acceleration is obtained. Hence, the vibration frequency can be estimated by the spectrum analysis of estimated vibration acceleration. With the estimated vibration acceleration and vibration frequency, the SFM signal is reconstructed. Then, the corresponding THz-SAR imaging algorithm is proposed to estimate and compensate the phase error caused by the high-frequency vibration error of the motion platform and realize high-frequency vibration error estimation and compensation for THz-SAR imaging. Finally, the effectiveness of the novel algorithm proposed in this paper is demonstrated by simulation results.

## 1. Introduction

Terahertz waves usually refers to electromagnetic waves from 100 GHz to 10 THz, which lie between the microwave and infrared, known as the THz gap. However, recent research results show that the THz gap has great development potential and application value due to its unique electromagnetic wave band [1,2]. In recent years, THz waves have been widely used in explosives detection, drug detection, imaging, radar and wireless broadband communication [3,4,5,6,7].

Compared with conventional microwave SAR imaging, THz-SAR imaging has considerable advantages in two-dimensional or three-dimensional ultrahigh-resolution imaging and detection and the recognition of weak-small-slow maneuvering targets due to its higher frequency and shorter wavelength. Ultrahigh range resolution can be obtained with a large bandwidth signal, and ultrahigh azimuth resolution can be obtained with a large Doppler bandwidth. In practical engineering application, airborne SAR systems are usually influenced by atmospheric disturbance, which introduces motion errors in radar echo signals. Low-frequency motion errors can be measured by motion sensors and compensated for in imaging processing. Due to the small amplitude, the influence of high-frequency vibration error on microwave SAR imaging is generally negligible. However, a higher frequency also means that the imaging quality of THz-SAR is more sensitive to the vibration error of the motion platform. Although the high-frequency vibration error will not affect the track of THz-SAR, it will seriously influence the phase of the radar echo signal, thus worsening the image quality [8,9]. Therefore, high-frequency vibration error estimation and compensation is the key for THz-SAR ultrahigh resolution imaging.

Since the high-frequency vibration error of a motion platform can be modeled as simple harmonic motion, the phase of THz-SAR-received echo signal in each range gate is modulated periodically. Meanwhile, the received echo signal of THz-SAR in each range gate is considered as sinusoid frequency modulation (SFM). Different from low-frequency motion error, high-frequency vibration error may produce a paired echo due to periodic modulation of the echo signal phase, which seriously affects the image quality. When the vibration frequency is very high, the traditional motion compensation algorithms will fail [10,11]. In [12], a THz-SAR imaging method under helicopter platform vibration was proposed. The compensation function is constructed by modern signal processing methods such as short-time Fourier transform and parametric spatial projection after pair echo is focused by adopting Keystone transform. Due to the non-stationary characteristics of SFM signals, the commonly used time-frequency analysis method cannot effectively analyze SFM signal. In [13], an SFM signal parameter estimation algorithm based on discrete sinusoidal frequency modulation transformation was proposed. It needs to search in three-dimensional space, resulting in huge computational burden. In [14], the simulated annealing algorithm was used to reduce the computational load, but converging to a local minimum occasionally is inevitable when it is used in practice, which leads to a large error in the parameter estimation results. In [15,16], sinusoidal frequency modulation Fourier transform and sinusoidal frequency modulation Fourier-Bessel transform were proposed to estimate SFM signal parameters, respectively. However, both these two algorithms directly make use of the SFM signal phase, which causes the algorithm performance to be seriously affected by signal-to-noise ratio (SNR) due to the phase unwrapping step.

Fractional Fourier transform (FrFT) is considered as a transform of signal from time domain into fractional Fourier domain (FrFD) between time and frequency [17]. As the FrFT of a signal is regarded as the decomposition in terms of an orthonormal basis set formed by chirp signals in FrFD, FrFT is a more suitable analysis tool for non-stationary signal processing. However, FrFT could not be directly used for the detection and parameter estimation of SFM signals. This is due to the fact that the phase order of SFM signals is more than two over the entire sample period and FrFT of SFM signals cannot have a good energy concentration property. Therefore, as the short-time Fourier transform (STFT) can determine the time–frequency relation of a time-varying signal, performing FrFT on SFM signals in a small-time window can determine the time–chirp rate relation. Through searching matched order in each sliding window, the relation of time–chirp rate of SFM signals can be obtained. Therefore, a novel algorithm for the parameter estimation of an SFM signal based on local FrFT (LFrFT) is proposed. The instantaneous chirp rate of SFM signal is determined by the matched order of LFrFT in sliding small-time window and the vibration acceleration is obtained. Hence, the vibration frequency is estimated by the spectrum analysis of vibration acceleration. With the estimated vibration acceleration and vibration frequency, the SFM signal is reconstructed and the high-frequency vibration error is obtained. Then, the corresponding THz-SAR imaging algorithm is proposed to estimate and compensate the phase error caused by high-frequency vibration error of motion platform and realize THz-SAR high-frequency vibration error estimation and compensation imaging. Finally, simulation experiments are made to verify the effectiveness of high-frequency vibration error estimation method and THz-SAR imaging algorithm with high-frequency vibration error.

This paper is organized as follows. THz-SAR imaging model with a high-frequency vibration error is established in Section 2. In Section 3 and Section 4, the LFrFT-based high-frequency vibration estimation method and the THz-SAR high-frequency vibration error compensation imaging algorithm are proposed, respectively. In Section 5, simulation experimental results are given to demonstrate the effectiveness of the proposed algorithm. Finally, a conclusion to this paper is drawn in Section 6.

## 2. THz-SAR Imaging Model with High-Frequency Vibration Error

In airborne SAR systems, the radar platform is usually influenced by atmospheric disturbance, which introduces motion errors in the radar echo signal. Among them, high-frequency vibration error is generally referred to meet the following conditions:(1)$$\left|f_{v}\cdot T_{s}\right|\geq 1$$
where $f_{v}$ is the frequency of high-frequency vibration error and $T_{s}$ is the synthetic aperture time of the imaging target.

The high-frequency vibration error of aircraft platform is close to a superposition of multiple harmonic motions with an octave relationship. The platform vibration amplitude of a helicopter is the composition of each order’s harmonic. However, the vibration amplitude of the first harmonic plays a leading role in the platform vibration [18]. Thus, only the first harmonic vibration is considered here. The high-frequency vibration error can be expressed as
(2)$$d\left(t\right)=A_{v}sin\left(2\pi f_{v}t+\varphi_{v}\right)$$
where $A_{v}$ is the amplitude of high-frequency vibration error and $\varphi_{v}$ is the initial phase of high-frequency vibration error.

Considering that the influence of high-frequency vibration error perpendicular to the line of sight on the imaging quality of THz-SAR can be ignored [19], only high-frequency vibration error in the plane along the line of sight is considered here, as shown in Figure 1. The high-frequency vibration error is in the YOZ plane, and the angle between the vibration direction and the Z axis is ϕ. The period of vibration is $T_{v}=1/f_{v}$. The radar platform flies horizontally along the X direction with a speed v. The height of the radar platform is H. The coordinates of the radar platform at the azimuth moment t can be expressed as (vt,−dsinϕ,H+dcosϕ). For arbitrary point target $P\left(vt_{0},y_{0},0\right)$ in the imaging region, whose zero doppler time is $t_{0}$, the instantaneous slant range R(t) can be expressed as
(3)$$\begin{array}{c} R\left(t\right)=\sqrt{v^{2}{\left(t-t_{0}\right)}^{2}+{\left(y_{0}+dsin\phi \right)}^{2}+{\left(H+dcos\phi \right)}^{2}}\\ \;\;\;\;\;\;\;\;\;\;\;\;\;\;\;=\sqrt{r_{s}^{2}\left(t\right)+d^{2}+2\left(Hcos\phi +y_{0}sin\phi \right)d} \end{array}$$
where $r_{s}\left(t\right)$ is the instantaneous slant range of the point target with the ideal motion state of radar platform and can be expressed as
(4)$$r_{s}\left(t\right)=\sqrt{v^{2}{\left(t-t_{0}\right)}^{2}+y_{0}^{2}+H^{2}}=\sqrt{v^{2}{\left(t-t_{0}\right)}^{2}+r_{0}^{2}}$$
where $r_{0}$ is the nearest slant range of the point target.

Since the synthetic aperture length of THz-SAR is short and the amplitude of high-frequency vibration error is far less than the nearest slant range $r_{0}$, (3) can be approximated as
(5)$$R\left(t\right)\approx r_{0}+\frac{v^{2}{\left(t-t_{0}\right)}^{2}}{2r_{0}}+\frac{\left(Hcos\phi +y_{0}sin\phi \right)}{r_{s}\left(t\right)}d\approx r_{0}+\frac{v^{2}{\left(t-t_{0}\right)}^{2}}{2r_{0}}+Cd$$
where $C=cos\left(\theta_{0}-\phi \right)<1$ and is constant, and $\theta_{0}=arctan\left(\frac{y_{0}}{H}\right)$ is the view angle of the point target.

Suppose that radar antenna transmits a linear frequency modulation (LFM) signal, whose center frequency is $f_{c}$, pulse duration time is T, and chirp rate is $K_{r}$. After mixing demodulation, the echo signal of point target received by radar antenna can be expressed as
(6)$$\begin{array}{c} s\left(\tau ,t;r_{0},t_{0}\right)=w_{r}\left(\tau -\frac{2R\left(t\right)}{c}\right)w_{a}\left(t-t_{0}\right)\\ \;\;\;\;\;\;\;\;\;\;\;\;\;\;\;\;\;\;\;\;\;\;\;\;\;\;\;\;\;\;\;\;\;\;\;\;\;\;\;\;\;\;\;\;\;\;\;\;\;\;\;\cdot exp\left\{-j\frac{4\pi }{\lambda }R\left(t\right)\right\}\cdot exp\left\{j\pi K_{r}{\left(\tau -\frac{2R\left(t\right)}{c}\right)}^{2}\right\} \end{array}$$
where c is the speed of light, λ is the carrier wavelength, and $w_{r}$ and $w_{a}$ are the antenna pattern of range direction and azimuth direction, respectively.

After range compression, it can be obtained from (5) and (6)
(7)$$\begin{array}{c} s\left(\tau ,t;r_{0},t_{0}\right)=sinc\left(\pi B\left(\tau -\frac{2r_{0}}{c}-\frac{v^{2}{\left(t-t_{0}\right)}^{2}}{cr_{0}}-\frac{2Cd}{c}\right)\right)w_{a}\left(t-t_{0}\right)\\ exp\left\{-j\frac{4\pi }{\lambda }\left[r_{0}+\frac{v^{2}{\left(t-t_{0}\right)}^{2}}{2r_{0}}+Cd\right]\right\} \end{array}$$
where B is the signal bandwidth.

The amplitude of high-frequency vibration error is very small, usually in the millimeter or sub-millimeter level, and the range resolution of THz-SAR is above the centimeter level. As the parameters listed in the simulation experiment, the vibration error amplitude is 0.5 mm and less than 1/8 of the range resolution, 6.64 cm. Therefore, the range cell migration caused by high-frequency vibration error can be ignored. After performing the range cell migration correction through interpolation in the range-doppler domain, (7) can be expressed as
(8)$$s\left(\tau ,t;r_{0},t_{0}\right)=sinc\left(\pi B\left(\tau -\frac{2r_{0}}{c}\right)\right)w_{a}\left(t-t_{0}\right)\cdot exp\left\{-j\frac{4\pi }{\lambda }\left[r_{0}+\frac{v^{2}{\left(t-t_{0}\right)}^{2}}{2r_{0}}+Cd\right]\right\}$$

After the dechirp operation and residual video phase (RVP) compensation [20], (8) can be expressed as
(9)$$s\left(\tau ,t;r_{0},t_{0}\right)=sinc\left(\pi B\left(\tau -\frac{2r_{0}}{c}\right)\right)w_{a}\left(t-t_{0}\right)\cdot exp\left\{-j\left[\frac{4\pi }{\lambda }r_{0}-2\pi \frac{2v^{2}t_{0}^{2}}{\lambda r_{0}}+\frac{4\pi }{\lambda }Cd\right]\right\}$$

For (9), the spectral analysis algorithm using the fast Fourier transform (FFT) operation can focus the azimuth signal in the doppler domain. However, due to the existence of high-frequency vibration error, the focused image will generate paired echoes, leading to a decrease in the image quality. If the parameters of the high-frequency vibration error are known, the effect can be eliminated by the phase compensation algorithm, thus ensuring the imaging quality of THz-SAR.

## 3. LFrFT-Based High-Frequency Vibration Error Estimation Method

According to (9), the phase error of an echo signal caused by high-frequency vibration error in each range gate can be regarded as the SFM signal, which can be expressed as
(10)$$g\left(t\right)=Aexp\left\{-j2\pi \frac{2}{\lambda }C_{1}sin\left(2\pi f_{v}t+\varphi_{v}\right)\right\}$$
where $C_{1}=CA_{v}$ is constant.

Suppose that $d_{LOS}\left(t\right)$ is the high-frequency vibration error along the line of sight, we can obtain that
(11)$$d_{LOS}\left(t\right)=Cd=C_{1}sin\left(2\pi f_{v}t+\varphi_{v}\right)$$

On the one hand, if we take the second derivative of $d_{LOS}\left(t\right)$ directly, the vibration acceleration along the line-of-sight direction is
(12)$$a\left(t\right)=\frac{\partial^{2}d_{LOS}\left(t\right)}{\partial t^{2}}=-4\pi^{2}f_{v}^{2}C_{1}sin\left(2\pi f_{v}t+\varphi_{v}\right)=-4\pi^{2}f_{v}^{2}d_{LOS}\left(t\right)$$

Therefore, if the vibration acceleration $\hat{a}\left(t\right)$ and vibration frequency ${\hat{f}}_{v}$ along the line-of-sight direction are both obtained, the high-frequency vibration ${\hat{d}}_{LOS}\left(t\right)$ along the line-of-sight direction can be estimated by
(13)$${\hat{d}}_{LOS}\left(t\right)=-\frac{\hat{a}\left(t\right)}{4\pi^{2}{\hat{f}}_{v}^{2}}$$

On the other hand, in a short time $\left[t_{0},t_{0}+\Delta t\right]$, $d_{LOS}\left(t\right)$ can be approximated by quadratic polynomials, namely
(14)$$d_{LOS}\left(t\right)=x\left(t_{0}\right)+v\left(t_{0}\right)t+\frac{1}{2}a\left(t_{0}\right)t^{2}$$
where $x\left(t_{0}\right)$ is the initial vibration displacement, $v\left(t_{0}\right)$ is the initial vibration velocity and $a\left(t_{0}\right)$ is the vibration acceleration.

Therefore, in the short time $\left[t_{0},t_{0}+\Delta t\right]$, (10) can be expressed as
(15)$$g\left(t\right)=Aexp\left\{-j2\pi \left[\frac{2}{\lambda }x\left(t_{0}\right)+\frac{2}{\lambda }v\left(t_{0}\right)t+\frac{1}{\lambda }a\left(t_{0}\right)t^{2}\right]\right\}$$

From (15), we see that it is an LFM signal. If the chirp rate $\hat{c}\left(t_{0}\right)$ has been obtained, the vibration acceleration $a\left(t_{0}\right)$ can be estimated through the following equation.
(16)$$\hat{a}\left(t_{0}\right)=-\frac{\lambda }{2}\hat{c}\left(t_{0}\right)$$

According to the definition of FrFT, the FrFT of the LFM signal y(t) with matched order $p_{0}$ is an impulse function at the fractional domain frequency $u_{0}$. At the same time, the matched order $p_{0}$ and the fractional domain frequency $u_{0}$ can be estimated by solving the following optimal equation [21], i.e.,
(17)$$\left\{{\hat{p}}_{0},{\hat{u}}_{0}\right\}=\arg\;\underset{p,u}{max}\left|F^{p}\left\{y\left(t\right)\right\}\right|$$
where $F^{p}\left\{y\left(t\right)\right\}$ is the *p*th FrFT of the signal y(t). Therefore, the chirp rate $c\left(t_{0}\right)$ can be determined by the matched order $p_{0}$, i.e.,
(18)$$\hat{c}\left(t_{0}\right)=-cot\left(\frac{\pi }{2}{\hat{p}}_{0}\right)$$

By selecting the starting time $t_{0}$ continuously and performing FrFT on the azimuth signal in the corresponding time interval $\left[t_{0},t_{0}+\Delta t\right]$ to estimate the local vibration acceleration $a\left(t_{0}\right)$, the vibration acceleration $\hat{a}\left(t\right)$ of the whole azimuth signal can be obtained. At the same time, it can be seen from (12) that the frequency of vibration acceleration along the line-of-sight direction is also $f_{v}$, and the vibration frequency ${\hat{f}}_{v}$ can be estimated through spectral analysis of the estimated vibration acceleration $\hat{a}\left(t\right)$. Since the FrFT is carried out in the local region of azimuth signal, we call it the local FrFT, denoted as LFrFT.

To sum up, the procedure of the LFrFT-based high-frequency vibration error estimation method is as follows:

(a) select the appropriate time interval Δt and divide the signal g(t) into N segments;

(b) for each segment signal, the grading iterative search method [22] is used to estimate the matched order ${\hat{p}}_{0}$ through (17);

(c) local vibration acceleration $\hat{a}\left(t_{0}\right)$ can be estimated by matched order ${\hat{p}}_{0}$ through (16) and (18);

(d) repeat steps (b) and (c) until the vibration acceleration of all segments’ signal is estimated;

(e) adopting the moving average method to filter the vibration acceleration to improve the estimation accuracy;

(f) perform FFT on the vibration acceleration and estimate the vibration frequency ${\hat{f}}_{v}$ from the spectrum;

(g) the estimation of high-frequency vibration error $\hat{d}\left(t\right)$ can be obtained through (11) and (12).

## 4. High-Frequency Vibration Error Compensation Imaging Algorithm

It can be seen from (9) that conventional imaging algorithms such as range Doppler (RD) and chirp scaling (CS) will generate paired echoes in SAR images and reduce the image quality when high-frequency vibration error exists in THz-SAR platform. Therefore, combined with the high-frequency vibration estimation method based on LFrFT, the high-frequency vibration compensation imaging algorithm of THz-SAR is proposed as follows:

Step 1. After range compression and range cell migration correction, the echo signal can be expressed as
(19)$$\begin{array}{c} s\left(\tau ,t;r_{0},t_{0}\right)=sinc\left(\pi B\left(\tau -\frac{2r_{0}}{c}\right)\right)w_{a}\left(t-t_{0}\right)\\ \;\;\;\;\;\;\;\;\;\;\;\;\;\;\;\;\;\;\;\;\;\;\;\;\;\;\;\;\;\;\;\;\;\;\;\;\;\;\;\;\;\;\;\;\;\;\;\;\;\;\;\cdot exp\left\{-j\frac{4\pi }{\lambda }\left[r_{0}+\frac{v^{2}{\left(t-t_{0}\right)}^{2}}{2r_{0}}+CA_{v}sin\left(2\pi f_{v}t+\varphi_{v}\right)\right]\right\} \end{array}$$

Step 2. Select the range gate with the dominant scatter, multiply it by the reference function $s_{ref}$ shown in (19) to complete the dechirp operation, and carry out RVP compensation, we can obtain the SFM signal shown in (10);
(20)$$s_{ref}=exp\left\{j\frac{4\pi }{\lambda }\left[r_{0}+\frac{v^{2}t^{2}}{2r_{0}}\right]\right\}$$

Step 3. Perform the LFrFT-based estimation method proposed in Section 3 and obtain the high-frequency vibration error;

Step 4. Adopt the estimated high-frequency vibration error to reconstruct the compensation signal $s_{com}$ and multiply it by (18) to complete the high-frequency vibration error compensation;
(21)$$s_{com}=exp\left\{-j\frac{4\pi }{\lambda }CAsin\left(2\pi f_{v}t+\varphi_{v}\right)\right\}$$

Step 5. Azimuth compression is done through matched filtering or dechirp operation, and the well-focused image is obtained.

Therefore, the flowchart of THz-SAR high-frequency vibration error estimation and compensation imaging algorithm is shown in Figure 2.

## 5. Simulation Experiments

In the section, the high-frequency vibration error estimation algorithm based on LFrFT is used to estimate the high-frequency vibration error, and then the new proposed THz-SAR high-frequency vibration compensation imaging algorithm is adopted for focusing on the imaging of a point target and an area target. The THz-SAR system’s simulation parameters are given in Table 1.

### 5.1. Simulation Analysis of LFrFT-Based High-Frequency Vibration Error Estimation

Considering the high-frequency vibration case with the form of (10) and supposing that, $f_{v}=20\;\mathrm{Hz},\;C_{1}=5\times 10^{-4}\;m$ and $\varphi_{v}=0$. It can be seen from Table 1 that the synthetic aperture time of the THz-SAR system is 0.4 s. The high-frequency vibration acceleration estimated by LFrFT and its spectrum are shown in Figure 3 and Figure 4, respectively. From the comparison between the estimated vibration acceleration and the true vibration acceleration, it can be found that the estimation error of vibration acceleration is larger at the extreme value points. This is due to the fact that the approximation error is larger when the second order approximation is adopted. At the same time, the estimated values of vibration acceleration after moving the average filter is smoother and closer to the true value of vibration acceleration than that of vibration acceleration before filtering. It is obvious from the spectrum diagram that the frequency of vibration acceleration is 20.04 Hz, which is almost consistent with the actual vibration frequency. Therefore, combining the estimated vibration acceleration and vibration frequency, the estimated vibration displacement can be obtained, as shown in Figure 5. According to the comparison between the estimated value and the true value of displacement, the LFrFT-based high-frequency vibration estimation method can estimate the SFM signal well.

In the following, we compare the performance of the proposed method with that of discrete sinusoid frequency modulation transform (DSFMT)-based method in [14]. One metric to evaluate the estimation performance is the normalized root-mean-square error (NRMSE), as in [15]. It is defined as
(22)$$\mathrm{NRMSE}=\frac{\left\|d_{estimate}\left(t\right)-d_{true}\left(t\right)\right\|}{\left\|d_{true}\left(n\right)\right\|}$$
where ‖⋅‖ denotes the $\ell_{2}$-norm operator. The NRMSE values the normalized Euclidean distance between the estimated high-frequency vibration error and the true one.

In the simulation experiment, the SNR of the signal is varied in 5 dB increments between 0 and 15 dB. For each SNR value, we apply the LFrFT-based method and DSFMT-based method to obtain the high-frequency vibration error, respectively, and 100 simulations are performed. Table 2 shows the average NRMSE of estimated high-frequency vibration error for LFrFT-based method and DSFMT-based method. From Table 2, we can see that the estimation accuracy degrades slowly with the decrease in the SNR, and the estimate performance of our proposed method is better than that of DSFMT-based method.

### 5.2. Simulation Analysis of High-Frequency Vibration Error Estimation and Compensation Imaging Algorithm

The THz-SAR system parameters in Table 1 and the high-frequency vibration error in simulation 5.1 are used to generate the echo signal for a single point target. Firstly, the RD algorithm is used for focusing on imaging, as shown in Figure 6a. It can be seen that the single point target has many paired echoes in the case of high-frequency vibration error. Then, the proposed high-frequency vibration error estimation and compensation imaging algorithm is adopted for focusing on imaging, as shown in Figure 6b. It is obvious that the point target is well focused and paired echoes is eliminated after the high-frequency vibration error is estimated and compensated.

In order to test the focus quality of the point target more intuitively, a rectangular window is adopted to intercept the point target in Figure 6b. After up-sampling eight times, the resulting two-dimensional sections (amplitude and phase) are shown in Figure 7, respectively. As can be seen from the two profiles, the range resolution and the azimuth resolution are 0.0664 and 0.0696 m, the peak sidelobe ratios are −13.2643 and −11.8592 dB, and the integral sidelobe ratios are −30.2416 dB and −25.6668 dB, respectively. The theoretical range resolution and azimuth resolution are known to be 0.0664 and 0.0692 m, respectively. Hence, the point target is well focused. Since the estimation of high-frequency vibration error is not ideal, the residual error after compensation makes the azimuth peak sidelobe ratio slightly higher. Therefore, it is necessary to further improve the estimation accuracy of the high-frequency vibration error or adopt the window function for suppressing the sidelobe.

Then, the multi-point targets are used for the simulation experiment. The THz-SAR imaging results of the RD algorithm and the new proposed imaging algorithm in this paper are showed in Figure 8a,b, respectively. It can be seen from Figure 8 that the new proposed imaging algorithm in this paper is still applicable to the case of multiple point targets and can effectively suppress paired echoes.

Finally, since there are no measured data of THz-SAR, the area targets are simulated to generate echo data in the experiment. The THz-SAR imaging results of the RD algorithm and the new proposed imaging algorithm in this paper are showed in Figure 9b,c, respectively. It can be seen that the new proposed imaging algorithm in this paper can effectively suppress paired echoes. However, the image quality after high-frequency vibration error compensation is slightly worse than the raw image shown in Figure 9a due to the residual error.

Remarks: The estimation accuracy of vibration displacement directly affects the focusing quality of THz-SAR image and is determined by the estimation accuracy of vibration acceleration with the LFrFT-based method. However, due to the existence of many factors, such as the signal-to-clutter-noise ratio of selected dominant scatter signal and window size of LFrFT, the estimation of high-frequency vibration displacement is not ideal, as shown in Table 2. After compensation with estimated high-frequency vibration displacement, the paired echoes in azimuth are suppressed in the THz-SAR image, but the image quality may still be a little defocused due to the residual error, as shown in Figure 9. In actual THz-SAR image processing, the residual phase error can be further extracted and compensated by using existing autofocus algorithms such as PGA.

## 6. Conclusions

Considering that THz-SAR image quality is sensitive to high-frequency vibration error of the motion platform, a novel THz-SAR high-frequency vibration error estimation and compensation imaging algorithm based on LFrFT is proposed in this paper. The THz-SAR echo signal received in each range pixel is modeled as SFM signal. The instantaneous chirp rate of SFM signal is estimated by the matched order of LFrFT in sliding small-time window and the vibration acceleration is obtained. Meanwhile, the vibration frequency is estimated by the spectrum analysis of vibration acceleration. With the estimated vibration acceleration and vibration frequency, high-frequency vibration error of motion platform is reconstructed. Then, the corresponding THz-SAR imaging algorithm is proposed to compensate the phase error caused by high-frequency vibration error of motion platform and realize THz-SAR high-frequency vibration error estimation and compensation imaging. Finally, the effectiveness of LFrFT-based high-frequency vibration error estimation and compensation imaging algorithm is verified by simulation experiments.

Although this paper proposes a corresponding solution to the problem of high-frequency vibration error for THz-SAR, the solution is also applicable to other wavebands, such as the millimeter band. In fact, the high-frequency vibration error of THz-SAR platform may contain a variety of vibration frequency components. Therefore, it is necessary to further study the high-frequency vibration compensation imaging algorithm of THz-SAR in the case of multi-frequency vibration error. The premise of the proposed method is that there are dominant scatter points in the imaging scene. The case where there is no dominant scatter point is also worth studying in future work.

## Figures and Tables

**Figure 1 sensors-20-02669-f001:**
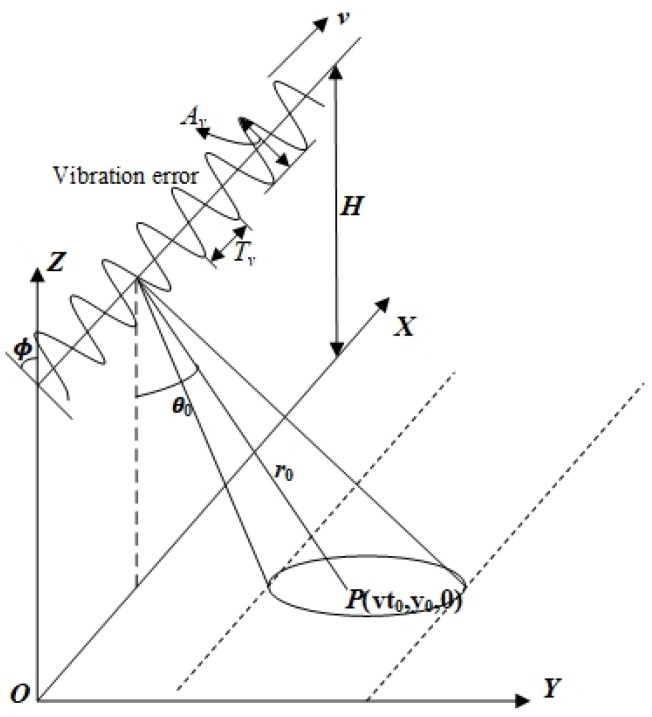
Terahertz synthetic aperture radar (THz-SAR) imaging geometry with high-frequency vibration error.

**Figure 2 sensors-20-02669-f002:**
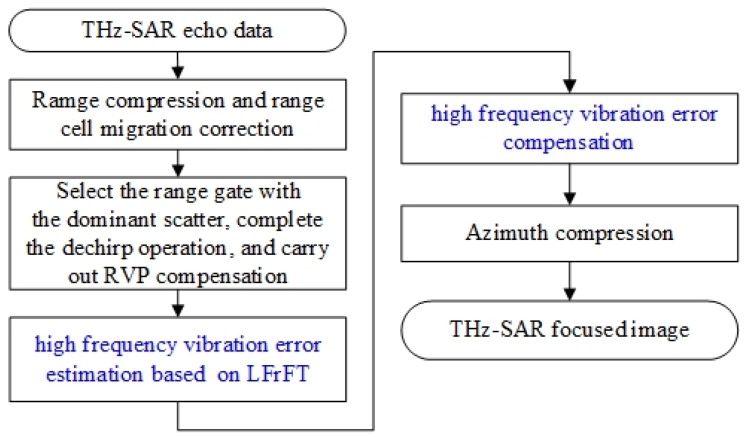
THz-SAR high-frequency vibration compensation imaging algorithm based on local fractional Fourier transform (LFrFT).

**Figure 3 sensors-20-02669-f003:**
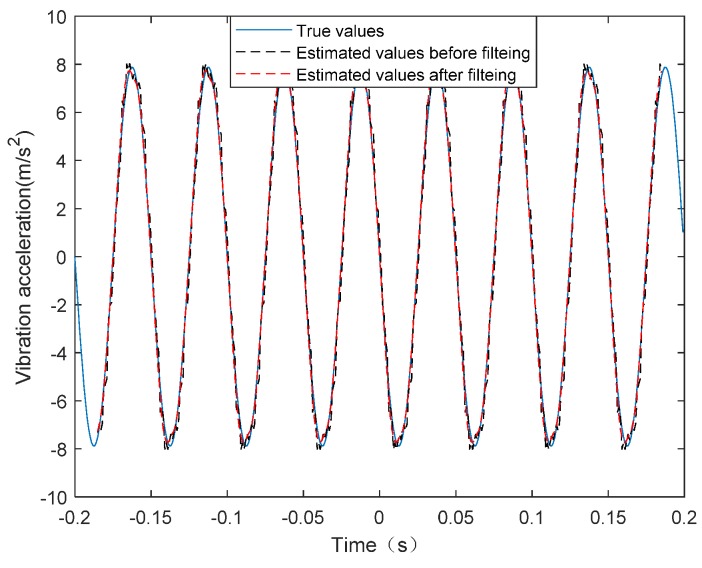
Real and estimated values of high-frequency vibration acceleration.

**Figure 4 sensors-20-02669-f004:**
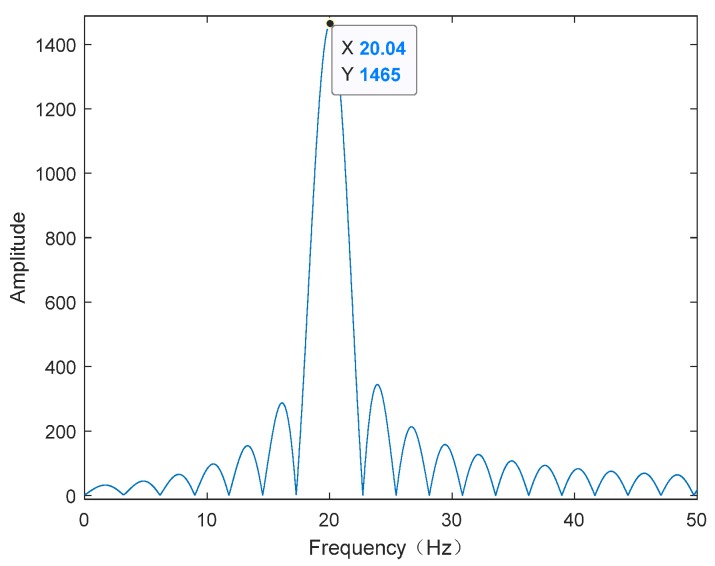
Frequency spectrum of high-frequency vibration acceleration.

**Figure 5 sensors-20-02669-f005:**
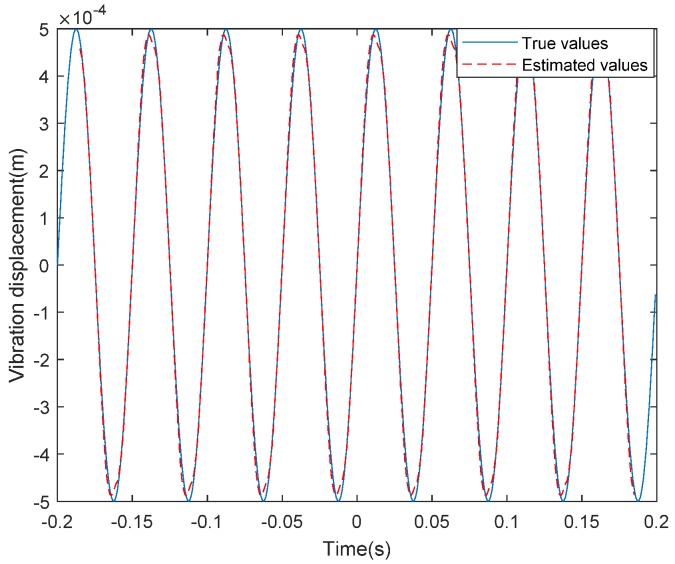
Real and estimated values of high-frequency vibration displacement.

**Figure 6 sensors-20-02669-f006:**
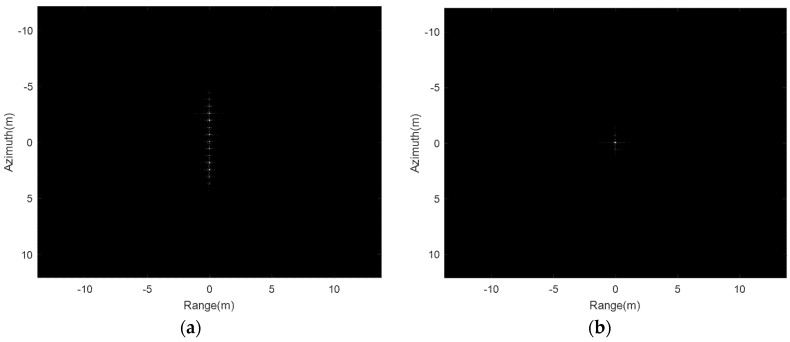
Point target image focused by (**a**) RD algorithm; (**b**) the proposed algorithm.

**Figure 7 sensors-20-02669-f007:**
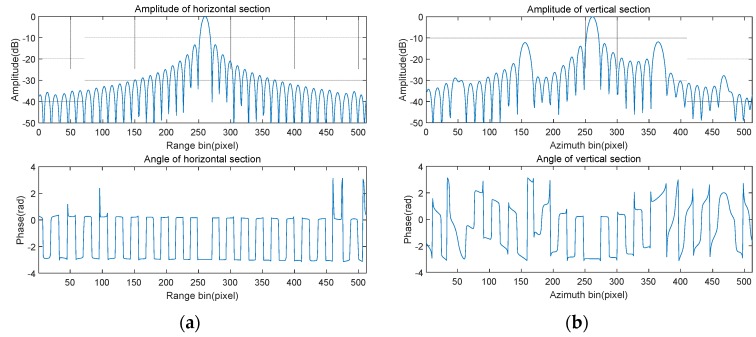
Focusing quality analysis of point target. (**a**) Range; (**b**) azimuth.

**Figure 8 sensors-20-02669-f008:**
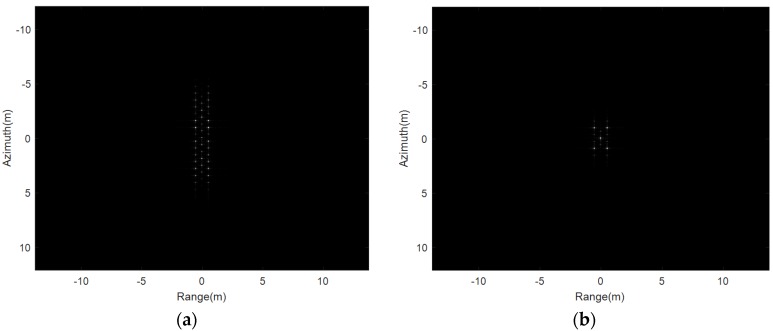
Imaging results of multi-point targets. (**a**) RD algorithm; (**b**) the propose algorithm.

**Figure 9 sensors-20-02669-f009:**
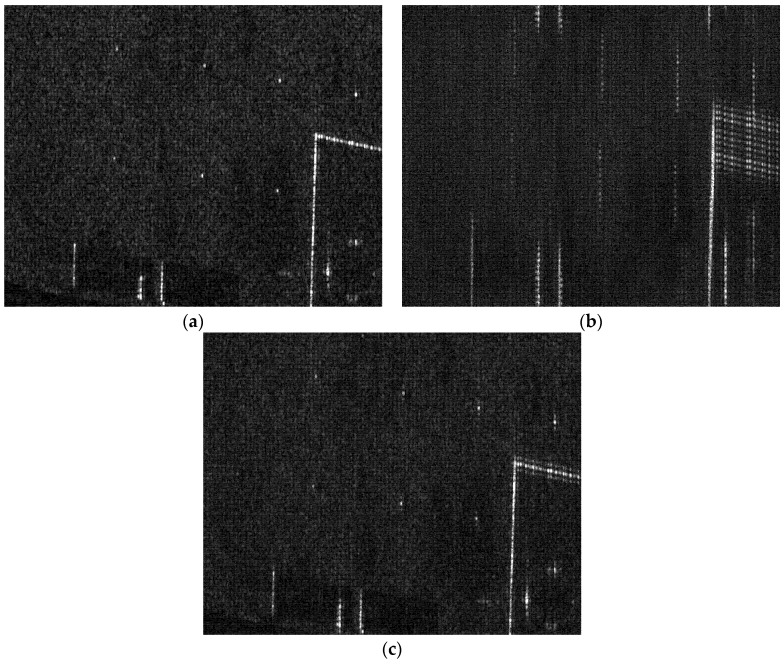
Imaging results of area targets. (**a**) Raw Image; (**b**) RD algorithm; (**c**) the proposed algorithm.

**Table 1 sensors-20-02669-t001:** System parameter table.

Parameter	Value
Center frequency	200 GHz
Platform height	2000 m
Platform speed	50 m/s
View angle	30°
Pulse width	1.5 µs
Signal bandwidth	2.0 GHz
Sample frequency	2.5 GHz
Pulse repetition frequency	1000 Hz

**Table 2 sensors-20-02669-t002:** Average normalized root-mean-square error (NRMSE) of estimated high-frequency vibration error under typical signal-to-noise ratios (SNRs).

SNR	Average NRMSE of DSFMT	Average NRMSE of LFrFT
0 dB	0.2285	0.1973
5 dB	0.1771	0.1234
10 dB	0.0829	0.0678
15 dB	0.0578	0.0352

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
