# Peer review of "A Novel High-Frequency Vibration Error Estimation and Compensation Algorithm for THz-SAR Imaging Based on Local FrFT"

_sensors, 2020, doi:10.3390/s20092669_

Round 1

Reviewer 1 Report

Authors proposed a parameter estimation method of SFM signals based on Local fractional Fourier transform. As a good topic in estimating the vibration errors of the airborne radar platform, the simulation results support their analysis.

(1)However, I expect the authors can possibly show more experiment results using the measured data. 

(2)Please modify flowchart in Fig. 2 to make the image clear.

(3)Can authors give an answer which may affect the results more, the center frequency or the bandwidth? I wonder if your method can be applied in other frequency band like mmWave band as I cannot find particular relevance to THz. 

Reviewer 2 Report

The authors worked on an algorithm based on Local Fractional FT for parameter error estimation and compensation for THz-SAR imaging. The paper needs a review for grammar and typos. The topic is relevant, but the paper has some points that is not clearly presented and must be improved. The variables and/or information from (2) as well as from lines 113 to 116 must be added in Fig. 1. The Fig. 1 does not have enough information. It must be clear for a first-time reader. The authors statement in lines 113-115 and lines 140-142 are contradictory, since the error in the vibration plane is modeled as a single sine in (2) and then it is written that it is small and son ignored by the authors. Comments or more information must be provided. For this reviewer, the paper misses a comparison with other publication/method as well as a quantitative assessment of the results. The plots and image are not sufficient to evaluate the performance of the algorithm. The authors should add error estimation, for instance. Another point to be considered is how do the results in Fig. 3 and Fig. 5 affect the image? Figs. 6 and 8 are important, but cannot be used for analysis or evaluation, since they are useful for visualization only. And how did the compensation algorithm work to minimize the error in THz-SAR imaging?

Reviewer 3 Report

The authors address the problem of high frequency vibrations in SAR Imaging in THz domain. The authors propose first a modelisation of the vibration as a sinusoidal frequency modulation which is, in my oppinion, a good model for such problems. Then, they propose a technique based on Local Fractional Fourier Transform to  estimate the effect of the vibrations and then to compensate the motion and improve the qualitfy of SAR image.

The mathematical formulation is very well described and the proposed technique is clearly presented.

My questions concern the results in realistic contexts (the simulation results reveal only very simple configuration, quite far from the real cases). What happens if the vibrations happen according to different directions and if they have many components?

In the same context, how the multi-component signals, usually common in radar Imaging, can be processed with proposed techniques?

I think the paper should address more complicate signals to prove the interest of the proposed technique.

Round 2

Reviewer 1 Report

1. Check the unit of phase in FIg. 7.

2. Give some reasons why measured experiment data is unavailable in this paper to further support that your simulation is enough.

3. The alignment of some parts in the paper should need proof-reading.

Reviewer 2 Report

The authors answered all questions and made the appropriate comments. No more concerns from this reviewer. However, some minor typos showed up in new authors’ edit. They are the following:

Line 109: write “Thus, only the first […], instead of “so only the first […];

Line 110: write “The high frequency […], instead of “And, high frequency […];

Lines 146, 291, 292, 293, and all other lines: separate the unit from the value – for example, write 0.5 mm, instead of 0.5mm.
